# Deep Learning-Based Detection of Cognitive Impairment from Passive Smartphone Sensing with Routine-Aware Augmentation and Demographic Personalization

Yufei Shen[1], Ji Hwan Park[1], Minchao Huang[1],
Jared F. Benge[2], Justin F. Rousseau[3,4], Rosemary A. Lester-Smith[5], and Edison Thomaz[1]

*Abstract*—**Early detection of cognitive impairment is critical for timely diagnosis and intervention, yet infrequent clinical assessments often lack the sensitivity and temporal resolution to capture subtle cognitive declines in older adults. Passive smartphone sensing has emerged as a promising approach for naturalistic and continuous cognitive monitoring. Building on this potential, we implemented a Long Short-Term Memory (LSTM) model to detect cognitive impairment from sequences of daily behavioral features, derived from multimodal sensing data collected in an ongoing one-year study of older adults. Our key contributions are two techniques to enhance model generalizability across participants: (1) *routine-aware augmentation*, which generates synthetic sequences by replacing each day with behaviorally similar alternatives, and (2) *demographic personalization*, which reweights training samples to emphasize those from individuals demographically similar to the test participant. Evaluated on 6-month data from 36 older adults, these techniques jointly improved the Area Under the Precision-Recall Curve (AUPRC) of the model trained on sensing and demographic features from 0.637 to 0.766, highlighting the potential of scalable monitoring of cognitive impairment in aging populations with passive sensing.**

*Index Terms*—**Digital Phenotyping, Mobile Sensing, Cognitive Impairment, Time Series Modeling, Personalization**

## I. INTRODUCTION

Cognitive decline associated with aging can significantly impair processing speed, working memory, and executive function, thereby reducing quality of life [1]. Early detection of cognitive dysfunction is crucial for timely diagnosis and intervention. However, clinical assessment instruments such as the Montreal Cognitive Assessment (MoCA) [2] are typically administered infrequently, failing to capture fluctuations influenced by mood, fatigue, medication, or environment, and potentially painting an incomplete or misleading representation of cognitive status. Participants may also alter their behaviors during assessments due to the Hawthorne effect, potentially biasing the results [3].

The widespread use of smartphones and wearables into daily life has emerged as a promising approach for health monitoring by passively capturing naturalistic human behaviors through sensor data from these devices. This technique, referred to as digital phenotyping [4], has attracted increasing attention for its potential to support scalable and longitudinal health tracking without requiring active user engagement. In the context of cognitive impairment, passive sensing has been used to investigate associations between cognitive function and various behavioral domains, such as physical activity [5], on-screen typing [6], and social engagement [7].

While prior studies have yielded meaningful insights into cognitive decline, most focused on a single or limited set of data modalities and only conducted statistical analyses. Some studies recorded multimodal signals across diverse behavioral dimensions and employed machine learning models to detect cognitive impairment [8]–[10]. In these studies, features were aggregated from sensor data over multi-week windows and classical models (e.g., Random Forest, XGBoost) were employed. A shortcoming of this approach is that feature aggregation may dilute informative signals and overlook fine-grained patterns embedded in the raw data collected at higher temporal resolutions.

Compared to classical models, Recurrent Neural Networks (RNNs), and especially LSTMs, are well suited for predicting health outcomes from behavioral sequences over longer and finely-grained temporal scales [11]–[13]. In this work, we implemented an LSTM model to detect cognitive impairment from sequences of daily behavioral features, derived from multimodal passive smartphone sensing data collected in an ongoing one-year study of older adults. To address the limited sample size, a key challenge in digital phenotyping research [14], we introduced two techniques to improve model generalizability:

- **Routine-Aware Augmentation**, which expands the training data by generating synthetic sequences in which each day is replaced with behaviorally similar alternatives.
- **Demographic Personalization**, which re-weights training samples to emphasize those from individuals demo-

[1]Department of Electrical and Computer Engineering, Cockrell School of Engineering, The University of Texas at Austin, Austin, TX, USA

[2]Department of Neurology, Dell Medical School, The University of Texas at Austin, Austin, TX, USA

[3]Department of Neurology, University of Texas Southwestern Medical Center, Dallas, TX, USA

[4]Peter O'Donnell Jr. Brain Institute, University of Texas Southwestern Medical Center, Dallas, TX, USA

[5]Department of Speech, Language, and Hearing Sciences, Moody College of Communication, The University of Texas at Austin, Austin, TX, USA

graphically similar to the test subject.

We systematically evaluated the model and techniques on 6-month data from 36 participants. These techniques jointly increased the AUPRC of the model trained on sensing and demographic features from 0.637 to 0.766 under the leave-one-participant-out (LOPO) cross-validation scheme.

## II. RELATED WORKS

### A. Digital Phenotyping for Cognitive Impairment

Digital phenotyping studies have investigated multidimensional behavioral signatures of cognitive impairment. To illustrate, Park [6] analyzed smartphone typing dynamics and found that longer keystroke hold times and transition times between consecutive keypresses were associated with poorer cognitive performance. Muurling et al. [7] characterized social engagement from phone calls, app usage, and location data. They found that cognitively impaired individuals exhibited more repetitive social behaviors, specifically calling the same contacts more frequently. A large-scale longitudinal study [15] tracked over 20,000 participants for two years using smartphones and wearables, with preliminary findings supporting the feasibility of detecting cognitive impairment through smartphone-based interactive assessments. Furthermore, the RADAR-AD study [9] developed machine learning models to differentiate stages of cognitive decline using various smartphone- and wearable-based remote monitoring technologies. Similarly, Chen et al. [8] trained XGBoost classifiers to detect cognitive impairment from 12 weeks of multimodal sensing data. Our work builds upon these efforts by leveraging deep learning to model fine-grained behavioral patterns across diverse domains for characterizing cognitive impairment.

### B. Deep Learning for Time Series in Health Sensing

Compared to classical machine learning models, deep learning approaches, such as RNNs, can capture nuanced temporal dynamics and behavioral patterns from frequently sampled sensing data. Among them, LSTM has been widely used due to its lightweight architecture and competitive performance in time series modeling. For instance, Hong et al. [12] used an LSTM to predict cognitive impairment from daily sleep variables collected via wearables over several months. Umematsu et al. [11] and Yu et al. [16] built LSTMs to forecast future wellbeing of college students based on time series derived from smartphone and wearable sensing. More recent efforts have explored large language models (LLMs) to infer wellbeing from behavioral time series [17], [18]. While these approaches show promise, modeling the complex behavioral phenotypes of cognitive decline [19] can be more challenging.

### C. Data Augmentation and Model Personalization

Data augmentation is a widely used technique to increase training data size and enhance the performance of deep learning models. Um et al. [20] applied various signal transformations to augment wearable accelerometer time series for monitoring Parkinson's disease. In contrast, our augmentation strategy operates on daily behavioral features rather than raw sensor signals. Specifically, we leverage participants' daily routines to generate synthetic trajectories by replacing each day with behaviorally similar alternatives.

Personalization improves model performance by tailoring it to individual participants. A common strategy is to introduce a portion of the test participant's data into the training set [21]. Yu et al. [16] further fine-tuned a participant-independent model using a small amount of data from the test subject for wellbeing prediction. While effective, these approaches violate subject-level independence and undermine LOPO evaluation's goal of assessing model generalizability to unseen individuals. Moreover, they require access to ground truth health outcomes for the test subject, posing challenges for cognitive impairment detection. Whereas wellbeing scores can be conveniently obtained via surveys or Ecological Momentary Assessments (EMAs), determining cognitive status requires time-consuming formal assessments. Therefore, models intended for scalable cognitive impairment detection should avoid relying on ground truth labels from the test participant. An alternative approach trains models on a subset of participants similar to the test subject based on personalization metrics (e.g., demographics and mental health scores) [13]. However, this reduces the amount of training data, which may be suboptimal for studies with relatively small cohorts.

To address these limitations in detecting cognitive impairment, our personalization strategy leverages instance weighting to emphasize training samples from participants with demographic profiles similar to the test subject. This approach preserves subject-level independence and utilizes all available training data.

## III. DATA ACQUISITION AND PROCESSING

### A. Study Protocol

Our one-year prospective observational study recruits community-dwelling older adults aged 65 years and above. At enrollment, participants provided informed consent and installed a custom-built smartphone app for passive sensing. Cognitive assessments from Version 3 of the Uniform Data Set by the National Alzheimer's Coordinating Center [22] were administered remotely every 6 months to evaluate participants' cognitive performance at baseline, 6 months, and 12-month study exit. Demographically adjusted assessment results were analyzed by a neuropsychologist in the study team to determine whether participants exhibited cognitive impairment. As of May 2025, the study is still actively recruiting and monitoring current participants. This manuscript focuses on the baseline cognitive performance of participants enrolled between May 2023 and December 2024.

### B. Smartphone Sensing Application

For data collection, we developed an iOS smartphone application that continuously captures multimodal data in the background without requiring any active user interaction. The app utilizes various iOS frameworks to record inertial measurement unit (IMU) readings, infer physical activities, track step counts, sense geolocations, and retrieve metrics

from iPhone's built-in Health app. In particular, it leverages the iOS SensorKit framework, only available to research studies reviewed and approved by Apple, to collect detailed smartphone interaction data while preserving user privacy. These interactions include smartphone and app usage, keyboard typing dynamics, and metadata from phone calls and text messages. The app transmits collected data to a secure remote server when the phone is connected to Wi-Fi and is either charging or has at least 50% of battery remaining.

### C. Passive Sensing Features

From the raw sensor data, we extracted 147 features to comprehensively characterize participants' daily behaviors, organized into 6 major categories described below. We first inferred participants' timezones from their location data and partitioned the raw data into daily data frames. Behavioral features of each day were then computed from these data frames. As some participants traveled during the study period, we excluded all days with multiple inferred timezones to avoid biasing the daily activity estimates.

*1) Activity:* The iOS Core Motion framework recognizes activities including walking, running, cycling, and automotive travel every few seconds. From these activity inferences, we summarized the total daily duration of each activity to capture participants' overall activeness.

*2) Pedometer and Gait:* We extracted both high-level and granular features from the iPhone pedometer data. Daily total step count and walking distance were computed to quantify overall activity levels, while we used the time of day when the first step was taken to reflect the timing of physical movement. To characterize participants' walking patterns in detail, we used the step timestamps to identify continuous walking periods of at least 10 seconds with more than 10 steps taken, and calculated statistics for the step count, distance, cadence (steps/second), and pace (seconds/meter) across all such periods during each day. The statistics, including the mean, selected percentiles (5th, 25th, 50th, 75th, and 95th), and median absolute deviation, provided robust representations of the feature distributions.

Furthermore, we obtained the daily minimum, average, and maximum of several gait metrics from the built-in Health app, including walking speed, step length, asymmetry, and double support time. These features complemented the statistics derived from continuous walking periods to capture more nuanced aspects of naturalistic walking. Specifically, walking asymmetry measures the proportion of steps with asymmetric speeds, and double support time represents the percentage of the gait cycle with both feet on the ground [23].

*3) Location:* To preserve privacy, raw location coordinates were shifted to obfuscate participants' true positions. Following established practices in location feature extraction [24]–[26], we excluded low-quality samples recorded under unreliable signal conditions, and classified the remaining ones as either stationary or moving. Specifically, samples with an accuracy over 100 meters or an instantaneous speed exceeding 180 km/h were removed. A sample was considered stationary if its maximum distance to any other sample recorded within a 10-minute window was less than 200 meters.

From these samples, we computed measures to quantify various aspects of participants' daily movement. Spatial variability was assessed using location variance, defined as the logarithm of the sum of variances in latitude and longitude [26]. Spatial extent was characterized by the total distance traveled and geometric properties of the convex hull, the smallest polygon enclosing all recorded locations, including its area, perimeter, and Gravelius compactness [27]. To capture temporal characteristics, we extracted stationary and moving durations, along with the earliest time of movement.

Furthermore, we assessed movement patterns with respect to the significant places participants visited. These places were identified by clustering stationary samples with the DBSCAN algorithm [28]. The cluster with the longest total stay between midnight and 6 a.m. was designated as the home location. To characterize general mobility patterns, we extracted the number of clusters and the time spent across all clusters and specifically at home. We also computed the maximum distance between any pair of clusters, as well as between home and other clusters, to capture spatial relationships among significant locations. The radius of gyration, defined as the average deviation of each cluster from the centroid of all clusters [29], was used to quantify spatial dispersion. Lastly, we calculated location entropy [26] based on the distribution of time spent across clusters, and extracted the time of day when participants were farthest from home to capture temporal aspects of their trajectories.

*4) Smartphone and App Usage:* We first extracted the total number of unlocks and unlock duration to assess overall smartphone usage. To protect user privacy, SensorKit did not record the names of third-party iOS apps, but logged the usage time for each of 29 predefined app categories (e.g., games, news, lifestyle). We consolidated these categories into 6 broader types: productivity, information, social, life, health, and other, and computed the proportion of usage time for each type to reflect detailed usage patterns.

*5) Typing:* SensorKit did not log any content typed by users. Instead, it recorded metadata from typing events and keystrokes. To reduce variability introduced by keyboard layout, we excluded all typing sessions in landscape orientation. We then extracted total typing duration and numbers of typing sessions and typed words as aggregate measures of overall typing activity. Additionally, we computed the frequency of various typing events, such as taps, deletes, altered words, corrections, and pauses, relative to the word count to reflect participants' typing dynamics.

Beyond these aggregate features, we derived keystroke-level metrics potentially indicative of fine motor control and cognitive function. Specifically, we extracted the hold time of character keys and estimated typing speed using the transition time between consecutive character inputs. We also obtained the transition time between character keys and deletes to capture self-correction behaviors. Typing accuracy was quantified by the spatial distance between each

character keystroke and the center of the corresponding key. To construct interpretable daily features, we applied the same set of summary statistics used in pedometer feature extraction to aggregate these keystroke-level measurements.

*6) Communication:* As a privacy safeguard, SensorKit does not collect the actual content of phone calls or text messages, nor any identifiable information about contacts (e.g., names or phone numbers). Therefore, we summarized the number of incoming and outgoing calls and text messages, total call duration, and the number of unique contacts involved in these communications to examine participants' social engagement.

## IV. Experimental Setup

### A. Dataset Preparation

Our goal was to develop a deep learning model to detect cognitive impairment based on participants' behavioral trajectories derived from passive sensing. Similar to prior study [8], window slicing was used to capture diverse temporal patterns while reducing variability from short-term events (e.g., travel). Specifically, we applied a 30-day sliding window to construct sequences of daily behavioral features, and advanced the window by one day to maximize the number of available sequences. Participant-level estimates were then obtained by averaging probability predictions across all sequences from each participant. To ensure the features accurately reflected daily behavior, we defined a valid day as one with at least 14 hours of sensing coverage between 6 a.m. and midnight. Sensing duration was also included in the feature set. Features were extracted only for valid days, and a sequence was retained if it contained at least 23 valid days. We also excluded participants with fewer than 5 sequences for robust predictions. Missing feature values were imputed as zero after standardization. To align with the timing of cognitive assessments, we focused on data collected during each participant's first 6 months of enrollment through March 2025. In total, we constructed 3,351 sequences covering 5,115 unique days from 36 participants, 12 of whom had cognitive impairment at baseline (age: 75.5 ± 5.2 years; education: 18.2 ± 1.5 years; 6 females) and contributed 981 sequences covering 1,595 days. The remaining 24 individuals were cognitively normal (age: 75.4 ± 5.4 years; education: 16.3 ± 1.9 years; 14 females) and contributed 2,370 sequences from 3,520 days.

### B. Classification Model

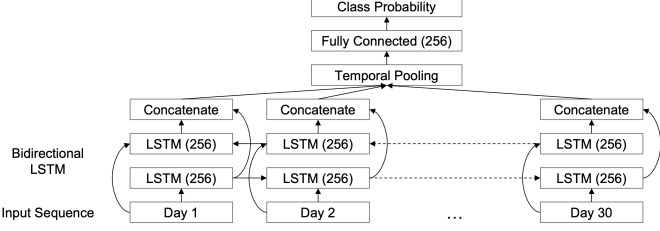

Fig. 1. Overall architecture of the LSTM model for detecting cognitive impairment from 30-day sequences of daily passive sensing features.

We used an LSTM for binary classification. As illustrated in Figure 1, it first processes the 30-day input sequence using a bidirectional LSTM layer with 256 hidden units to produce a 512-dimensional representation for each day. The daily representations are then averaged across the time axis to obtain a global representation of the entire sequence. This global vector is passed through a ReLU-activated fully connected layer with 256 units and 0.2 dropout. Finally, a classification head outputs the probability of cognitive impairment.

### C. Routine-Aware Augmentation

Our data augmentation strategy leverages participants' routines to generate synthetic day sequences in which each day is replaced with behaviorally similar alternatives. Specifically, for each pair of days $(i, j)$ from a participant, we computed the Euclidean distance $D_{ij}$ between their standardized sensing features vectors $\mathbf{x}_i, \mathbf{x}_j \in \mathbb{R}^d$: $D_{ij} = \sqrt{\sum_{k=1}^{d}(x_{i,k} - x_{j,k})^2}$. For each day $i$, we identified its 5 closest neighbors as replacement candidates $\{C_i\}$. To avoid substituting atypical days that deviate from routines with behaviorally dissimilar neighbors, only neighbors with distances below a threshold $\tau$ were retained. We set $\tau$ as the 10th percentile of all pairwise distances $\{D_{ij}|i < j\}$. Synthetic sequences were then generated by randomly sampling replacement days from $\{C_i\}$ for each day $i$ in the original sequence. Days without any valid replacements (i.e., no candidates with distances below $\tau$) or sufficient sensing coverage were left unchanged.

### D. Demographic Personalization

We developed a personalization method that preserves subject-level independence while utilizing data from all training participants. Specifically, it reweights training samples based on demographic similarities between training and test participants. Each participant was represented by a standardized three-dimensional demographic vector $\mathbf{d}$ from their age, sex, and years of education. We then computed Euclidean distances $S_{ij}$ between $\mathbf{d}_i$ of the test participant $i$ and $\mathbf{d}_j$ of each training participant $j$. All training samples from participant $j$ were assigned a weight $w_j$ using a softmax over the inverse distances to the test participant:

$$w_j = \frac{e^{1/S_{ij}}}{\sum_{k=1}^{M} e^{1/S_{ik}}} * N$$

where $M$ is the number of training participants and $N$ is the total number of training samples. This weighting scheme prioritizes training samples from participants demographically similar to the test subject while preserving the average weight of one across all samples to ensure comparability to uniform weighting. We further applied a softmax over the sample weights within each training batch to more effectively capture their relative importance.

### E. Experiments

We conducted a series of experiments to systematically evaluate the LSTM classifier and quantify the benefits of routine-aware augmentation and demographic personalization under a LOPO evaluation scheme. Model performance was assessed using both Area Under the ROC Curve (AUC) and Area Under the Precision-Recall Curve (AUPRC) for

comparability with prior study [8]. AUPRC emphasizes accurate predictions of the minority class and is therefore well suited for our imbalanced dataset, which includes fewer participants with cognitive impairment (i.e., the positive class).

As a demographic baseline, we fit a logistic regression on participants' age, sex, and years of education. An XGBoost model was trained on summary statistics (mean, SD, min, max) of the 147-dimensional passive sensing features computed over each 30-day sequence as a non-deep learning baseline. For the LSTM models, we optimized the balanced cross-entropy loss using an Adam optimizer with a learning rate of $5 \times 10^{-6}$ and a batch size of 128. To improve generalizability, label smoothing with a factor of 0.1 was applied. The base LSTM was trained for 30 epochs.

To evaluate the effect of routine-aware augmentation, we generated 5 synthetic sequences for each real sequence, increasing the training data size by 5 times. An LSTM model was then trained on the augmented dataset for 5 epochs to match the total number of optimization steps in the base setting for a fair comparison. We further trained an LSTM on the augmented dataset with demographic personalization to assess its additional contribution to model performance. In this case, the final loss of a batch was computed as the sum of balanced cross-entropy losses per included sample, each weighted by its personalization weight. To examine the impact of directly incorporating demographic context, all three LSTM settings were repeated on a fused feature set, where age, sex, and education were added as static inputs to each timestep of the passive sensing sequence.

We reported both sequence-level and participant-level performance for the XGBoost and LSTM models. The deterministic logistic regression was trained with a single random seed, while the others were trained with 10 different seeds. We used the same set of seeds across experiments to ensure fair comparison, and reported the mean ± SD across seeds as a robust estimate of model performance.

## V. RESULTS

### A. Overall Performance

Table I summarizes the classification performance across different combinations of feature sets and training settings. We used one-sided one-sample t-tests to compare model performance against the demographic baseline and one-sided paired t-tests to assess performance differences between other models. The models produced comparable results at the sequence and participant levels. At the participant level, the demographic baseline achieved an AUC of 0.656 and AUPRC of 0.473, both exceeding the expected performance of random guessing with 0.5 for AUC and 0.33 (i.e., prevalence of the positive class) for AUPRC.

The LSTM model trained on passive sensing features significantly outperformed the demographic and non-deep learning baselines in identifying participants with cognitive impairment, yielding an average AUPRC of 0.604. This demonstrates its effectiveness in modeling fine-grained behavioral trajectories. Routine-aware augmentation further increased its AUC from

0.660 to 0.671 and AUPRC from 0.604 to 0.623. More notably, demographic personalization led to a substantial performance gain, boosting AUC to 0.756 and AUPRC to 0.689. All improvements in AUC and AUPRC, from the baselines to LSTM, and with augmentation and personalization, are statistically significant ($p < .001$), except for the increase of AUC from the demographic baseline to LSTM ($p = 0.26$).

The benefits of augmentation and personalization were even more pronounced when sensing features were fused with demographic variables to train LSTMs. Augmentation improved participant-level AUC and AUPRC of the base model from 0.702 to 0.709 and from 0.637 to 0.654, respectively. Further personalization led to the best-performing model across all experiments, achieving an AUC of 0.780 and an AUPRC of 0.766. To put this result in context, Chen et al. [8] reported an AUPRC of 0.701 using XGBoost classifiers trained on combined sensing and demographic features. Our models that incorporated demographic information also outperformed their counterparts trained on sensing features alone, demonstrating the value of demographic context in detecting cognitive impairment. Again, all performance improvements reported here are statistically significant.

We further used the GradientExplainer from Shapley Additive Explanations (SHAP) [30] to identify important features utilized by the best-performing LSTM model for detecting cognitive impairment. Key contributors included higher education level, longer character key hold and transition times during typing (also reported in prior studies [6], [8]), more smartphone unlocks, and slower walking speed.

### B. Visualization of Participant Routines

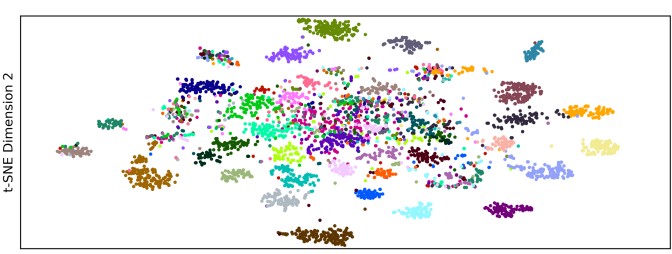

Fig. 2. t-SNE visualization of participants' daily passive sensing features from days with sufficient sensing coverage, color-coded by participant ID.

To visualize participants' daily routines, we obtained 4,384 unique days with sufficient sensing coverage from the 30-day sequences used in model development. Principal Component Analysis (PCA) was applied to the standardized daily features to retain 54 components that explained 95% of the total variance. We then used t-Distributed Stochastic Neighbor Embedding (t-SNE) [31] to project these components into a two-dimensional space. Figure 2 illustrates the resulting embeddings, color-coded by participant ID.

The visualization revealed clearly identifiable participant clusters, indicating the presence of routine behaviors across days. Specifically, many participants exhibited distinct routines, as reflected by their well-separated clusters. Others showed more similar behavioral patterns, with clusters located

LOPO PERFORMANCE ACROSS DIFFERENT COMBINATIONS OF MODELS, FEATURE SETS, AND TRAINING SETTINGS. *Aug* DENOTES ROUTINE-AWARE
AUGMENTATION, AND *Per* INDICATES DEMOGRAPHIC PERSONALIZATION. BEST VALUES FOR EACH METRIC ARE BOLDED.

| Model | Feature Set | Setting | AUC | | AUPRC | |
|---|---|---|---|---|---|---|
| | | | Sequences | Participants | Sequences | Participants |
| Logistic Regression | Demographics | Base | – | 0.656 | – | 0.473 |
| XGBoost | Sensing | Base | 0.518 ± 0.030 | 0.505 ± 0.034 | 0.331 ± 0.031 | 0.389 ± 0.037 |
| LSTM | Sensing | Base | 0.697 ± 0.011 | 0.660 ± 0.016 | 0.606 ± 0.014 | 0.604 ± 0.020 |
| | | Base + Aug | 0.701 ± 0.011 | 0.671 ± 0.015 | 0.612 ± 0.013 | 0.623 ± 0.021 |
| | | Base + Aug + Per | 0.814 ± 0.010 | 0.756 ± 0.010 | 0.727 ± 0.031 | 0.689 ± 0.026 |
| LSTM | Sensing + Demographics | Base | 0.735 ± 0.023 | 0.702 ± 0.025 | 0.603 ± 0.023 | 0.637 ± 0.025 |
| | | Base + Aug | 0.738 ± 0.024 | 0.709 ± 0.030 | 0.607 ± 0.026 | 0.654 ± 0.031 |
| | | Base + Aug + Per | **0.832 ± 0.016** | **0.780 ± 0.021** | **0.786 ± 0.033** | **0.766 ± 0.035** |

closer to each other near the center of the plot. Moreover, atypical days that deviated from routines appeared as outliers relative to their corresponding clusters. These observations justified the design of our routine-aware augmentation, which only replaced routine days with behaviorally similar alternatives when generating synthetic day sequences. They also provided empirical support for the effectiveness of this strategy in increasing the diversity of training data and enhancing model generalizability to unseen participants.

### C. Demographic Analysis

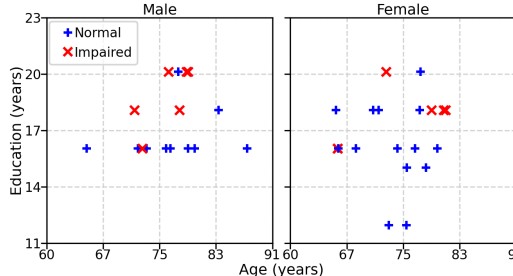

Fig. 3. Scatter plots of age and education for male and female participants, color-coded by cognitive status.

The two participant groups were roughly matched in age and gender, while those with cognitive impairment had approximately two more years of education on average. As reported in Section V-A, the demographic baseline outperformed random guessing in detecting cognitive impairment, and combining demographic variables with sensing features improved model performance. These findings suggest that demographic characteristics provide complementary information for detecting cognitive impairment.

To further explore potential mechanisms underlying the performance gains from demographic personalization, we visualized participants' age and education, stratified by sex and color-coded by cognitive status, in Figure 3. While no globally separable clusters were apparent, localized groupings were observed in which a few participants with the same cognitive status had similar demographic profiles. For example, three cognitively impaired female participants shared the same education level, with ages differing by less than 2 years. These observations indicate that our personalization strategy effectively leveraged demographic information by emphasizing behavioral patterns from individuals similar to the test participant.

As described in Section IV-D, the strategy employs a participant-level softmax and a batch-level softmax to derive sample weights from demographic similarity. In practice, we found it critical to have both components to achieve the substantial performance improvement reported. While removing either softmax retained more than half of the original gain in AUC, hardly any improvement was observed for AUPRC. This suggests that both demographic-based participant importance and the relevance of samples within each batch were effectively utilized through softmax normalization to adaptively prioritize more informative training samples, especially for identifying participants with cognitive impairment (i.e., the minority class).

## VI. DISCUSSION AND CONCLUSION

### A. Future Directions

We identified several directions for future research. First, this work used behavioral features aggregated at the day level. Building on this foundation, future work could examine behavioral trajectories at finer temporal scales. For example, app usage is summarized every 15 minutes, and physical activity is inferred every few seconds. Leveraging these higher-resolution time series may allow models to capture more nuanced behavioral signatures of cognitive decline. Second, we required sufficient sensing coverage within each day and across the 30-day windows to ensure reliable daily feature extraction. However, this criterion excluded several participants with inconsistent data collection. Notably, since smartphone use can be cognitively demanding, such inconsistencies may themselves carry information about cognitive function. Future research could explore event-based modeling approaches that do not rely on continuous sensing. For instance, pedometer and typing data can be analyzed at the event level (e.g., continuous walking periods or typing sessions), enabling model development from collections of discrete behavioral episodes. Lastly, it is essential to validate our modeling approach on both future participants from this ongoing study and independent external cohorts to establish its potential for real-world clinical deployment.

## B. Conclusion

In this work, we collected passive smartphone sensing data from older adults and extracted multimodal features to comprehensively characterize their daily behaviors. We then developed an LSTM classification model to detect cognitive impairment based on 30-day behavioral trajectories from 36 participants. To improve model generalizability and tailor it to individual-specific behavioral patterns, we introduced two strategies: routine-aware augmentation and demographic personalization. Evaluated with LOPO cross-validation, these techniques jointly increased the participant-level AUPRC from 0.604 to 0.689 for the LSTM trained on sensing features alone, and from 0.637 to 0.766 for the model trained on fused sensing and demographic features. Visualizations of participant routines and demographics provided additional empirical support for the effectiveness of the proposed strategies.

## ACKNOWLEDGMENT

This work is supported by NIH grant R01AG077017.

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
