# OpenReview forum: "Deep Learning-Based Detection of Cognitive Impairment from Passive Smartphone Sensing with Routine-Aware Augmentation and Demographic Personalization"
_IEEE.org/EMBS/BHI/2025/Conference — BHI 2025_

### Official Review · Reviewer_C9ew · 2025-07-14
**Deep Learning-Based Detection of Cognitive Impairment from Passive Smartphone Sensing with Routine-Aware Augmentation and Demographic Personalization**

**Confidence:** 4
**Clarity Of Writing:** great
**Clinical Significance:** good
**Methodological Novelty:** great
**Overall Rating:** 7

**Experiments And Results:**

great

**Questions For The Authors:**

1. Do you plan to validate the proposed model on an independent dataset or a larger cohort?

2. Given the relatively small dataset, what steps did you take to prevent overfitting beyond LOPO validation? Were any regularization techniques applied?

3. Is it possible that demographic weighting could introduce any unintended bias?

4. Do you anticipate any challenges in translating this model in real-world?

**Strengths:**

1. The paper addresses an important problem.
2. Implements routine-preserving augmentation to improve data diversity. This is novel and well-justified.
3. adds demographic personalization which could be helpful in improving model relevance.

**Summary Of The Paper:**

The paper presents a deep learning model to detect cognitive impairment using passive smartphone sensing data. The model uses time-series features: activity, location, communication, and is evaluated using leave-one-participant-out (LOPO) cross-validation on a small dataset of 36 participants.

**Weaknesses:**

1. Dataset is small.
2. No external validation.
3. The result could be potentially overfitted. The model is large compared to the dataset. No training or test loss curves have been shown. Also, regularization strategies are not mentioned or used.
4. Also, the feature collection (147 feature time-series) could be resource intensive.

---

### Official Review · Reviewer_Lsyt · 2025-07-15
**A well-explained study on detecting cognitive impairment using multimodal data with clear technical novelties.**

**Confidence:** 5
**Clarity Of Writing:** excellent
**Clinical Significance:** great
**Methodological Novelty:** great
**Overall Rating:** 7

**Experiments And Results:**

great

**Questions For The Authors:**

No.

**Strengths:**

1. The focused area has clinical significance
2. The dataset is solid, and the collection procedure and data details are well-explained.
3. Thoughtful technical contributions with strong experimental evidence.
4. The paper is well-written.

**Summary Of The Paper:**

This paper focuses on detecting cognitive impairment in older adults using passive smartphone sensing data. It introduces a dataset collected from 36 participants and proposes a deep learning model enhanced by two techniques: routine-aware data augmentation and demographic personalization. The paper compares the proposed method with baselines, and shows that the proposed deep learning model and two techniques all contribute to improved performance.

**Weaknesses:**

1. The paper would benefit from a deeper discussion on clinical integration and real-world deployment challenges.
2. The paper uses 6 categories of sensor data, but does not assess the individual contributions of these modalities to the classification task.
3. The 3D visualization in Figure 3 (demographic analysis) could be improved to provide clearer insights.

---

### Official Review · Reviewer_bJGP · 2025-07-16
**Review of "Deep Learning-Based Detection of Cognitive Impairment from Passive Smartphone Sensing with Routine-Aware Augmentation and Demographic Personalization"**

**Confidence:** 4
**Clarity Of Writing:** great
**Clinical Significance:** fair
**Methodological Novelty:** good
**Overall Rating:** 4
**Final Rating:** 5

**Experiments And Results:**

good

**Questions For The Authors:**

Could you provide an EDA to clarify which features or behavioral domains are most associated with cognitive impairment? For example, do certain routine disruptions consistently appear in the impaired group, and what specific patterns drive the model’s predictions?

The interpretation of Figure 3 is somewhat unclear. The axes are not labeled. I do see the two gender clusters, but it is difficult to agree with the claim that participants with similar cognitive status cluster together demographically. Consider adding axis labels and cluster boundaries, or maybe use another type of visualization.

**Strengths:**

The paper contributes a longitudinal clinical dataset, and potentially shows the feasibility of passive smartphone sensing for longitudinal cognitive health monitoring in older adults. The augmentation techniques are supported by thoughtful justifications and visualizations.

The experimental results in Table 1 clearly show that the LSTM model substantially outperforms the other two baseline models. There is also consistent improvement when adding demographic features and the two proposed training strategies. The evaluation set is small using LOPO, but AUPRC mean ± SD metric is reported and SD is small enough to show the model works consistently across participants.

**Summary Of The Paper:**

This paper contributes a clinical dataset of multimodal smartphone sensing data with corresponding baseline cognitive assessments from 36 older adults. The paper also presents a deep learning approach for cognitive impairment detection using a LSTM model that learns from sequences of daily behavioral features derived from the collected data. To address small-sample generalizability, the paper applied two novel strategies: routine-aware augmentation, which synthetically expands the training data by replacing daily behaviors with similar days; and demographic personalization, which reweights training samples to prioritize data from participants demographically similar to the test individual. An evaluation using leave-one-participant-out shows that these strategies significantly improve AUPRC.

**Weaknesses:**

The paper does not provide an analysis of which features contribute most to the binary classification result. This limits clinical interpretability and understanding. A feature importance analysis, at least for the original dataset without augmentation, would clarify this.

The binary cognitive status label (Normal/Impaired) is determined from clinical assessments administered every six months, which introduces a temporal mismatch with the 30-day sensing windows used as input. Related to (1), an EDA showing how specific features correlate with the cognitive labels would help validate the signal in the data.

---

### Official Review · Reviewer_dMfd · 2025-07-17
**Deep Learning-Based Detection of Cognitive Impairment from Passive Smartphone Sensing with Routine-Aware Augmentation and Demographic Personalization**

**Confidence:** 4
**Clarity Of Writing:** good
**Clinical Significance:** good
**Methodological Novelty:** fair
**Overall Rating:** 6

**Experiments And Results:**

good

**Questions For The Authors:**

- How many of those real sequences belong to each class.
- What are the LSTM hyper parameters?
- How the dataset is divided into training, validation, and test sets.

**Strengths:**

1.	Effective handling of class imbalance: The authors actively address the class imbalance by applying ‘Routine-Aware Augmentation’ to balance the dataset, enabling the model to treat each group more fairly.
2.	Appropriate evaluation metrics: They report model performance using AUC and AUPRC curves, which are more suitable than accuracy for evaluating imbalanced datasets.
3.	Bias mitigation through demographic personalization: They incorporate ‘Demographic Personalization’ to improve predictions for demographically similar individuals while mitigating bias caused by over- or under-represented groups in the training data.
4.	Appropriate model choice with superior results: The authors select an LSTM model, well-suited for sequential time-series data, and achieve results that surpass previous work in this domain.

**Summary Of The Paper:**

This research aims to detect signs of cognitive impairment (e.g., Alzheimer’s) using passively collected smartphone data. The authors leverage deep learning models and improve accuracy through routine-aware data augmentation and demographic personalization.

**Weaknesses:**

1.	Unclear class distribution: The authors do not specify the number of samples in the minority and majority classes, making it difficult to assess the level of imbalance.
2.	Synthetic data distribution is not explained: The authors mention generating “five synthetic sequences for each real sequence,” but fail to clarify how many of those real sequences belong to each class.
3.	Incomplete hyperparameter reporting: They do not provide a comprehensive list of LSTM hyperparameters (e.g., layers, hidden units, learning rate, dropout), which limits reproducibility.
4.	Missing dataset split information: The authors didn’t mention how the dataset is divided into training, validation, and test sets, making it unclear how they ensured proper generalization and avoided data leakage.
5.	Single dataset evaluation: To validate the robustness of ‘Routine-Aware Augmentation’ and ‘Demographic Personalization’, the authors should evaluate the model on multiple datasets.
6.	Lack of model explainability: Enhance interpretability by using Shapley values (SHAP) to visualize and quantify the contributions of major features to the model’s predictions.
7.	Inappropriate Handling of Missing Values: Missing values were imputed as zero, which is not a recommended approach.